# Obtaining a Monoclonal Antibody against a Novel Prometryn-Like Hapten and Characterization of Its Selectivity for Triazine Herbicides

**DOI:** 10.3390/bios13010022

**Published:** 2022-12-25

**Authors:** Lingyuan Xu, A. M. Abd El-Aty, Jing Zhao, Xingmei Lei, Xiuyuan Zhang, Yun Zhao, Xueyan Cui, Yongxin She, Fen Jin, Jing Wang, Maojun Jin, Bruce D. Hammock

**Affiliations:** 1Institute of Quality Standard and Testing Technology for Agro-Products, Chinese Academy of Agricultural Sciences, Beijing 100081, China; 2State Key Laboratory of Biobased Material and Green Papermaking, Qilu University of Technology, Shandong Academy of Sciences, Jinan 250353, China; 3Department of Pharmacology, Faculty of Veterinary Medicine, Cairo University, Giza 12211, Egypt; 4Department of Medical Pharmacology, Medical Faculty, Ataturk University, 25240 Erzurum, Turkey; 5College of Life Sciences, Yantai University, Yantai 264005, China; 6Department of Entomology & Nematology and the UC Davis Comprehensive Cancer Center, University of California, Davis, CA 95616, USA

**Keywords:** prometryn, hapten, monoclonal antibody, immunoassays

## Abstract

In this study, a previously unreported 3-((4-(isopropylamino)-6-(methylthio)-1,3,5-triazin-2-yl) amino) butyric acid hapten was designed and synthesized. This maximized the exposure of the antigen-determinant isopropyl of prometryn to the immune system in animals to induce the production of anticipated highly specific anti-prometryn antibodies. The hapten has a molecular weight of 285.37 Da. The compound was confirmed by nuclear magnetic resonance hydrogen spectroscopy (^1^H NMR), nuclear magnetic resonance carbon spectroscopy (^13^C NMR), and high-resolution mass spectrometry (HRMS). By using the active ester approach, immunogens and coated antigens were created. Bovine serum albumin (BSA) was used as an immunogen, along with the successfully produced hapten, to immunize mice. The IC_50_ value of mouse monoclonal anti-prometryn antibody (mAb) 7D4 (the quantity of analyte that generated 50% prometryn inhibition) was 3.9 ng/mL. The anti-prometryn mAb was of the IgG1 subclass. The IC_20_ (80% binding level (B/B_0_) of prometryn)-IC_80_ (20% binding level (B/B_0_) of prometryn) range of the anti-prometryn monoclonal antibody standard curve working range was 0.9–18.1 ng/mL. The prepared mAb has good characteristics because it can specifically recognize prometryn, and the cross-reaction rates for ametryn, desmetryn, and terbumeton were 34.77%, 18.09%, and 7.64%, respectively. The cross-reaction rate with the other seven triazines was less than 1%. The hapten structure proposed can serve as an additional tool for modulating selectivity in detecting triazines.

## 1. Introduction

Prometryn (2-methylthio-4,6-di(isopropylamino)-1,3,5-triazobenzene) is a triazine herbicide that has been widely or selectively used to control a variety of weeds in agricultural and nonagricultural fields worldwide [1]. It is stable in slightly acidic or alkaline ergometrine media and hydrolyzed in strong acid or base conditions [2]. Prometryn inhibits the interception of electrons transferred to oxidized coenzyme II (NADP^+^) by photosynthetic electrons, thus achieving the goal of weeding [3]. It is widely and excessively used in the environment, with a half-life of approximately 13 months [4]. These residues accumulate in soil and water, posing a serious threat to biological diversity and human health [5]. For example, using prometryn in cotton fields often negatively impacts cotton growth and reduces production [6,7]. It was previously reported that the detection rate of prometryn in some waters was as high as 12.5%. Triazine pesticides, such as atrazine, showed unacceptable carcinogenic risk estimates of 4.6 × 10^6^ [5]. Prometryn impacts water and plants when it enters water bodies. Research shows that prometryn has certain toxicity to seaweed and fish [8]. The effect of prometryn on human health is not only the direct effect of application [9] but also includes the indirect harm caused by the ingestion of contaminated products with high residues. Prometryn can impair people’s brain and immunological systems and lead to anomalies in reproductive organs [10,11]. As the impact of prometryn residues on the environment and human health has become increasingly prominent, various countries and regions have also strengthened the monitoring of prometryn dosage and set a limit for its maximum level. Despite being prohibited in Europe since 2004 [12], it is nonetheless commonly used in many other nations, including China, the US, Japan, and many more [11]. In China, the maximum residue limit (MRL) of prometryn in corn and fresh corn is 0.02 mg/kg, 0.05 mg/kg in rice, chestnut, brown rice, cottonseed, soybean, garlic, vegetable soybean, lotus seed (fresh), and lotus root is 0.05, and 0.1 mg/kg in peanut kernel and pumpkin. In December 2017, based on acute and chronic dietary analysis, carcinogenicity and exposure assessment of prometryn in the human body, the US Environmental Protection Agency (EPA) issued the final regulation to revise the MRL of prometryn in celery and lettuce, cotton seeds, sesame seeds, and fresh fennel leaves and stems to 0.50 mg/kg, 0.25 mg/kg, 0.50 mg/kg, and 0.05 mg/kg, respectively (https://regulations.justia.com/regulations/fedreg/2017/12/04/2017-26083.html, accessed on 4 December 2017). In the MRL formulated by Japan, rice (brown rice), wheat, and rye are 0.1 mg/kg. Corn (maize, combining popcorn and sweet corn), peanuts, (dried) yam, onion green, soybeans, and barley were 0.02 mg/kg. Therefore, the residual level needs to be monitored.

Gas chromatography (GC) [13], high-performance liquid chromatography (HPLC) [14], chromatography and mass spectrometry (GC-MS) [15,16], and capillary electrophoresis [17] are some of the established, tried-and-true methods for determining prometryn. Instrumental methods were considered to be highly sensitive and accurate. However, the instruments required for analysis are expensive. Furthermore, the loaded samples need complex preprocessing, cumbersome sample testing operations, and well-trained laboratory personnel, which cannot meet the requirements for rapid and extensive screening tests for onsite monitoring. Immunoassays based on antigen antibody-specific binding reactions are simple, fast, and accurate. The diversification of marker species has become one of the most valuable and potential trace analytical techniques for pesticides. Due to the agricultural product market size, daily test samples, and fresh characteristics of fruits and vegetables, it is urgent to adopt simple, accurate, sensitive, inexpensive, and rapid detection methods for screening pesticide residues in large quantities outdoors. The application potential is high for the production of antibodies with high specificity, sensitivity, and affinity. The primary starting point for developing immunoassay techniques is now the production of high-performance antibodies.

In this study, a novel prometryn-like hapten (3-((4-(isopropylamino)-6-(methylthio)-1,3,5-triazine-2-yl) amino) butyric acid) was proposed and synthesized. In this study, the chemical structure formula of the hapten was designed to expose the isopropyl group in the molecular structure formula of prometryn as an antigen determinant to the host immune system. The molecular weight and structure of the synthesized compound were identified. The synthesized compound has high purity and accurate structure. A complete antigen was prepared by coupling the hapten successfully synthesized with the carrier protein by the active ester method. Immunize mice with the successfully synthesized immunogen. A positive hybridoma cell line was screened, named mAb7D4, and the antibody type was IgG1. The hapten structure proposed can serve as an additional tool for modulating selectivity in detecting triazines. The purpose of this study was to prove that prometryn isopropyl was also an antigen determinant of triazine pesticides. This will be of great significance for the preparation of triazine antibodies. In addition, this study will provide additional effective core elements for the establishment of immunoassay methods for triazines.

## 2. Materials and Reagents

The triazine standard (prometryn, ametryn, desmetryn, terbumeton, propazine, terbuthylazine, simazine, simetryn, atrazine, prometon, and terbutryn) used by the Institute was secured from the first standard (Tianjin, China). Cyanuric chloride, tetrahydrofuran, isopropylamine, N,N-diisopropylethylamine (DIPEA), ethyl acetate, and sodium methyl mercaptan were purchased from Beijing Lianhe Technology Co., Ltd. (Beijing, China). Sodium chloride (NaCl), anhydrous sodium sulfate (Na_2_O_4_S), hydrochloric acid (HCl), sodium dihydrogen phosphate dodecahydrate (Na_2_HPO_4_·12H_2_O), sodium chloride (NaCl), gelatin, citric acid monohydrate, and Tween-20 were procured from Sinopharm Chemical Reagent Co., Ltd. (Beijing, China). Aladdin supplied anhydrous N,N-dimethylformamide (DMF). Sigma-Aldrich^®^ (Shanghai, China) Trading Co., Ltd. (Shanghai, China) supplied 1-ethyl-(3-dimethylaminopropyl) carbodiimide hydrochloride (EDC), N-hydroxysuccinimide (NHS), Freund’s complete adjuvant, Freund’s incomplete adjuvant, bovine serum albumin (BSA), and ovalbumin (OVA). Jackson Immunoresearch Laboratories Co. (The Cell Resource Center of Peking Union Medical College in Beijing, China) provided the mouse Sp2/0-Ag14 myeloma cell line. Thermo Fisher Scientific (Waltham, MA, USA) supplied the cell culture medium (GIBCO^®^ Australian Premium FBS (Waltham, MA, USA), GIBCO^®^ DMEM basic (1X) basal culture medium, penicillin solution, and L-glutamine solution). Costar^®^ (Corning, NY, USA) provided 96-well microplates and cell culture plates. IgG HRP (labeled goat anti-mouse IgG with HRP) was purchased from Jackson Immuno. Research Laboratories Inc. (West Grove, PA, USA).

## 3. Methods

### 3.1. Synthesis of Prometryn Hapten

The reaction of reaction solution 1 and reaction solution 2 was carried out in a 250 mL flask. 2,4,6-Trichloro-1,3,5-triazine (C_3_Cl_3_N_3,_ 7.37 g, 40 mmol) was dissolved in tetrahydrofuran (120 mL), and the solution was cooled to 0 °C as reaction solution 1. Isopropylamine (C_3_H_9_N, 2.36 g, 40 mmol) and N,N-diisopropylethylamine (DIPEA, 7.74 g, 60 mmol) were dissolved in tetrahydrofuran (50 mL) and used as reaction solution 2. Next, dropping solution 2 into solution 1 took approximately 15 min to complete. Then, the mixture was reacted at 0 °C for 3 h, and thin-layer chromatography (TLC) was used to detect whether the reaction was complete. After the reaction was completed, the mixture was spun dry, and ethyl acetate (100 mL) was added to the spin-dried beaker. The obtained mixture was washed twice with deionized water (100 mL) and once with saturated sodium chloride aqueous solution (100 mL). The mixture was dried with anhydrous sodium sulfate. Finally, it was dried to obtain a yellow oil mixture (SM1, 7.85 g) with a 94.8% yield.

4,6-Dichloro-N-isopropyl-1,3,5-triazin-2-amine (SM1, 10.8 g, 52.2 mmol) and sodium methyl mercaptan (3.6 g, 52 mmol) were dissolved in N,N-dimethylformamide (DMF, 100 mL) in a 100 mL flask. The reaction was carried out at room temperature (25 °C) for 16 h. TLC was used to monitor whether the SM1 reaction was completed. After the reaction, the reaction mixture was poured into water (600 mL). The reaction mixture was extracted twice with ethyl acetate (150 mL), and the organic phase was combined after extraction. The organic phase was washed three times with deionized water (150 mL) and once with saturated sodium chloride solution (100 mL). The organic phase solution obtained was dried with anhydrous sodium sulfate. The dried solution was spun dry to obtain 13.5 g of a yellow oil. The yellow oil was passed through a silica gel column (petroleum ether (PE): ethyl acetate (EtOAc) = 25:1) to obtain yellow oil (SM2, 9 g) with a yield of 79.6%. The purity of the obtained compound was determined to be 60% using liquid chromatography-mass spectrometry (LC-MS), which can be tested in the following experiment.

In a 250 mL flask, homemade 4-chloro-N-isopropyl-6-(methylthio)-1,3,5-triazin-2-amine (SM2, 9 g, 40 mmol), 3-aminobutanoic acid (SM3, 4 g, 40 mmol) and DIPEA (10.4 g, 80 mmol) were dissolved in ethanol (120 mL). The reaction mixture was heated to 90 °C and carried out for 8 h. TLC was used to monitor that the reaction stopped when approximately half of the raw material remained. The reaction mixture was cooled to room temperature (25 °C) and spun dry. A saturated sodium bicarbonate aqueous solution (50 mL) was added to the spin-dried reactant, and then the mixed solution was stirred for 20 min. The mixture after the reaction was extracted twice with ethyl acetate (100 mL). After extraction, the aqueous solution of the reaction liquid was collected, and the pH of the aqueous phase was adjusted to 4 with 3 M HCl. After pH adjustment, the solution was extracted three times with 100 mL dichloromethane. The extracted mixture was dried with anhydrous sodium sulfate. The dried mixture was spun dry to obtain 6.6 g of a yellow oil. Ethanol (50 mL) was added to the dried yellow oil, and the mixture was heated and refluxed for 1 h. After the reaction, the mixture was cooled to room temperature (25 °C). The white precipitate in the mixture was filtered and dried to 2.35 g.

### 3.2. Immunogen and Coating Antigen Preparation

When small molecule haptens of pesticides combine with large molecules (usually proteins), they have immunogenicity. This study used the active ester method to prepare coating antigens and immunogens. BSA and OVA were combined with proton hapten to make coating and immunogen antigens. Figure 1b depicts the immunogen and coating antigen synthesis pathways. The molar ratio of hapten to a carrier protein (BSA) used in the preparation of the immunogen was 60:1. The molar ratio of hapten to a carrier protein (OVA) used in the preparation of the coating antigen was 40:1. A total of 7.78 mg (0.027 mmol) of hapten, 6.28 mg (0.054 mmol) of N-hydroxysuccinimide (NHS), and 10.46 mg (0.054 mmol) of carbodiimide hydrochloride (EDC) were fully dissolved in 1 mL of anhydrous N,N-dimethylformamide (DMF) and reacted with magnetic stirring at 4 °C overnight (10 h). After the reaction, a reactant containing prometryn hapten activation was obtained, and the reaction product can be directly used for subsequent carrier protein coupling. The supernatant of 0.333 mL of hapten activation solution was added to the carrier protein OVA solution (the solution was obtained by dissolving 10 mg of OVA in 1 mL of 0.01 M phosphate buffer saline solution (PBS, pH = 7.4). The supernatant of 0.664 mL of hapten activation solution was added to the carrier protein BSA solution (the solution was obtained by dissolving 20 mg of BSA in 2 mL of 0.01 M PBS (pH = 7.4). The coupling reaction was stirred at 25 °C for 4 h. The reaction solution was dialyzed at 4 °C with a PBS solution with a pH value of 7.4 and a concentration of 0.01 mol/L. The dialysate was replaced every 4 h 6 times. After dialysis, the solution obtained from the reaction was quickly frozen in liquid nitrogen and stored at −20 °C. Matrix-assisted laser desorption/ionization time of flight mass spectrometry (MALDI-TOF-MS) was used to characterize the synthesized coating antigen and immunogen and calculate the coupling ratio. The coupling ratio was calculated as follows:Conjugation ratio = (*M_conjugate_* − *M*_carrier_)/*M_hapten_*.(1)
where the antigen conjugates are denoted by “*M_conjugate_*,” the BSA/OVA standard by “*M*_carrier_,” and the hapten by “*M_hapten_*.”

### 3.3. Production and Characteristics of Prometryn mAb

#### 3.3.1. Immunization

The Institute of Quality Standards and Testing Technology for Agro-Products’ Experimental Animal Welfare and Ethical Committee (IQSTAP-2021-05) approved the use of animals in experiments. The buffer solution (0.01 M PBS, pH = 7.4) was used to dilute the immunogen (2 mg/mL) to 1 mg/mL. After that, Freund’s adjuvant was added to an identical volume of 0.1 mL in a sterile glass vial. The mixture was placed on a magnetic stirrer and stirred and mixed rapidly in an ice bath to fully emulsify. Six 7-week-old female BALB/c mice were selected for immunization. At the beginning of the third immunization, 3–5 days after each immunization, blood was taken from the orbit, incubated at 37 °C for 0.5 h and 4 °C for 2 h, and centrifuged at 1000 r/min for 10 min. Serum was collected, and the serum titer was determined. The immune strategy of mice was the same as that previously reported [18]. Freund’s incomplete adjuvant was utilized for the first vaccination, and for the subsequent booster vaccination, Freund’s incomplete adjuvant was utilized. Freund’s adjuvant and immunogen during immunization should be 1:1 and thoroughly mixed until dripping into the water to prevent diffusion. The interval of booster immunization was 2 weeks, and the immune dose was 0.1 mg. Three days before fusion, the mice with the best performance were immunized with a dose of 0.05 mg. The mice were intraperitoneally injected without adjuvant. Appendix A displays the serum titer and inhibition rate of the mice with the highest performance. Appendix A displays the serum titer and inhibition rate of the selected fusion mice.

#### 3.3.2. Ic-ELISA Procedure

The ic-ELISA method was used to screen the best-performing fusion mouse sera. They were used to establish the standard curve of antibodies by detecting the supernatant of positive cells. The buffer required for ic-ELISA was the same as previously reported [18]. The specific steps of the operation are as follows: (1) The coated antigen was added to the microplate by diluting it with the required coating solutions (CBS, pH 9.6, 0.05 mol/L). Each well contained 100 µL, which was incubated at 37 °C for three h before being washed three times with washing buffer (PBST, PBS containing 0.1% Tween-20). (2) With sample dilution buffer (PBSTG, PBST containing 0.1% gelatin), the standard prometryn sample and the antiserum/antibody from mice were diluted to the required concentration. The microplate was incubated for 30 min at 37 °C with 50 µL of diluted antiserum or antibody and standard samples of various concentrations. PBST was used three times to wash the dishes. (3) Then, 100 µL of PBSTG-diluent IgG HRP (labeled goat anti-mouse IgG with HRP) was added to each well. After 30 min at 37 °C, the plate was cleaned three times with PBST. (4) Each well was incubated for 15 min in the dark at 25 °C with 100 µL of TMB color developing solution. To stop the reaction, 50 µL of the hydrochloric acid solution was added to each well. The absorbance at OD450 nm was measured with a TECAN Infinite M200 PRO microplate reader from Männedorf, Switzerland.

#### 3.3.3. Production of mAb

Ic-ELISA was used to measure the inhibition rate and serum titer of six immunized mice. Three days before cell fusion, Balb/c mice with the highest performance were selected and immunized. SP2/0 myeloma cells and mouse spleen cells were combined for fusion. After fusion, the supernatant of hybridoma cells was determined by indirect competitive enzyme-linked immunosorbent assay, the positive wells were screened, and the hybridoma cells were cloned to obtain hybridoma cell lines that can stably secrete monoclonal antibodies against prometryn. After the positive cell lines were expanded and cultured, mice treated with paraffin 7 days earlier were injected intraperitoneally. Ascites were collected after abdominal swelling and centrifuged at 4 °C at 1000 rpm, and the potency and inhibition rate of ascites were determined and frozen at −20 °C for purification. The 7D4 mAb against prometryn was obtained by purifying the culture medium with the octanoic acid-saturated ammonium sulfate method. The steps of antibody purification were as follows: (1) The pH was adjusted to 4.5 by diluting the antiserum or ascites (produced by hybridoma cell lines in the abdominal cavity of mice) with 60 mmol/pH 4 acetate buffer solution in a 1:4 ratio. Dropwise, 75 μL/mL serum was added, stirred for 30 min at room temperature, incubated for 1 h at 4 °C, and then centrifuged for 30 min at 10,000 rpm. (2) Qualitative filter paper was used to filter the sediment, and 0.1 mol/L PBS was added to the supernatant. A 1:10 (PBS: pH was adjusted to 7.4 with 1 mol/L NaOH, the supernatant) dilution ratio was added, and the sample was precooled for 15 min at 4 °C. (3) (NH_4_)_2_SO_4_ was added to a mixed solution containing 0.277 g/mL, stirred for 30 min, chilled for 2 h at 4 °C, centrifuged for 20 min at 12,000 rpm, and then discarded. (4) The precipitate was dialyzed four times with 0.01 mol/L PBS at 4 °C, pH 7.4, and twice with 0.01 M PB (0.2 g KH_2_PO_4_, 2.96 g Na_2_HPO_4_ • 12H_2_O, 1 L) after being dissolved in a small amount of 0.01 mol/L PBS. (5) Dry freeze and store at −20 °C until needed. (6) Safety Tips: If the time is short, it is best to extract immunoglobulin at 2–4 °C. This can also be done at room temperature, but all materials should be cooled before use. Ic-ELISA was used to measure the inhibition rate and titer of freeze-dried antibody powder. The procedure was identical to that of the antiserum.

## 4. Results and Discussion

### 4.1. The Results of Hapten, Immunogen, and Coating Antigen Identification

#### 4.1.1. Identification of Hapten

The obtained dried product was impure. High-performance liquid chromatography (HPLC) was used to purify the product, and 800 mg of oil was obtained in a 6% yield. Finally, the molecular weight and structure of the hapten were characterized by HRMS, ^1^H NMR, and ^13^C NMR. Hapten was confirmed by liquid chromatography, ^13^C NMR (Figure 2a), ^1^H NMR (Figure 2b), and HRMS (Figure 2c) as 3-((4-(isopropylamino)-6-(methylthio)-1,3,5-triazine-2-yl) amino) butyric acid.

The determination results of the chemical structural formula of prometryn hapten are shown in Figure 2d. The theoretically accurate molecular weight [M + ^+^H] of C_11_H_19_N_5_O_2_S was 285.37, which was 286.1398. The results showed that the expected hapten structure of prometryn was synthesized.

^1^H NMR (400 *MHz*, DMSO-d6) δ 7.55 (d, J = 8.2 *Hz*, 1H), 7.42 (s, 1H), 7.30 (s, 1H), 4.31 (dq, J = 14.2, 7.4 *Hz*, 2H), 4.10–3.99 (m, 2H), 2.57 (dd, J = 15.2, 6.4 *Hz*, 1H), 2.40 (s, 5H), 2.34 (dd, J = 14.7, 7.1 *Hz*, 1H), 1.13 (p, J = 7.5, 6.1 *Hz*, 16H).

^13^C NMR (101 M*Hz*, DMSO) δ 172.50, 43.26, 40.11, 39.90, 39.69, 39.48, 39.28, 39.07, 38.86, 22.27, 21.96, 20.16.

#### 4.1.2. Identification of the Prometryn Antigen and Immunogen

The molar ratio of prometryn used in the coating synthesis was 40:1, and the molar ratio of prometryn used in the immunogen synthesis was 60:1. The single charge ion peaks of the OVA standard and coated antigen detected by MALDI-TOF-MS were 44,586.160 and 46,144.036, respectively. The BSA standard and immune antigen had single charge ion peaks of 72,079.531 and 67,308.849, respectively. It was calculated from the coupling ratio formula of the carrier protein and hapten. The determined OVA standard and coating antigen, BSA standard, and immune antigen are shown by MALDI-TOF-MS in Figure 3.

### 4.2. Characterization and Cross-Reactivity of the 7D_4_ mAb

#### 4.2.1. Characterization of mAb

The ic-ELISA method was used to determine the ideal antibody and antigen mix. The coating antigen and antibody concentration used in the experiment was 1 mg/mL. The dilution multiples of the coating were 1 × 10^3^, 2 × 10^3^, 4 × 10^3^, and 8 × 10^3^. The prometryn standard was 1000 ng/mL. The best concentration combination of antigen and antibody was selected to establish the standard curve of the antibody. The determination results of the optimal combination are shown in Appendix A. Prometryn standards were diluted to 1000, 500, 250, 125, 62.5, 31.25, 15.625, 7.8125, 3.90625, and 0 ng/mL, with PBSTG for subsequent ic-ELISA experiments. The standard curve of the prometryn mAb 7D4 is shown in Figure 4. The standard curve was determined by ic-ELISA. The IC_50_ was the concentration of analyte that generated 50% prometryn inhibition. Each measured OD value in the curve is the average of three independent repeats. The 7D4 mAb’s inhibition rate was well linearly correlated with the prometryn concentration. The IC_50_ of the 7D4 mAb was 3.9 ng/mL, and the working range of the standard curve (IC_20_–IC_80_) was 0.9–18.1 ng/mL. The 7D4 mAb type was identified as IgG1 using a mouse antibody isotyping kit from Sigma^®^, St. Louis, MO, USA.

#### 4.2.2. Cross-Reactivities of the 7D_4_ mAb

Eleven triazine herbicides were selected to determine their cross-reactivity with 7D4 mAb. The determination and comparison results of the cross-reaction rate of 7D4 mAb are shown in Table 1. The cross-reaction rate of mAb prepared by the hybridoma cell line 7D4 screened in this study with prometryn was 100%, which indicated that the prepared 7D4 mAb could specifically recognize prometryn. The cross-reaction rates with ametryn, desmetryn, and terbumeton were 34.77%, 18.09%, and 7.64%, respectively. The cross-reaction rate with the other seven triazines was less than 1%. The hybridoma cell line 7D4 screened in this study can specifically recognize prometryn. However, this does not mean that the antibodies prepared by the positive hybridoma cell strains screened after immunizing host animals with hapten proposed in this study can specifically recognize prometryn. The cross-reaction rate may vary depending on the immunoassay format and the competitive hapten preparation used.

## 5. Conclusions

This study proposed a new structure design and synthesis route, which provides an additional tool for adjusting the selectivity in the detection of triazines. It is maximally exposed to the isopropyl group of the prometryn epitope to produce specific monoclonal antibodies against prometryn. The molecular weight of the prometryn hapten is 285.37 Da. The mAb 7D_4_ for the specific detection of prometryn was obtained, and its IC_50_ value was 3.9 ng/mL. The antibody type of 7D4 mAb was determined to be IgG1. The working range of the standard curve (IC_20_–IC_80_) was 0.9–18.1 ng/mL. The 7D4 mAb can be used as the core material for establishing prometryn immunoassays. The cross-reactivity values are not invariable parameters of a given antibody and may vary depending on the immunoassay format and the competing hapten preparation used, which provides additional tools for modulating selectivity in the detection of triazines.

This study once again confirmed that the isopropyl group of the structural formula of triazine pesticides is also a crucial antigenic determinant. We previously verified the hapten design of atrazine and concluded that the isopropyl group of triazine pesticides could be the epitope [18]. The hapten structure of triazine pesticides previously reported is shown in Table 2. Compared with previous reports on the hapten structure of prometryn [18,19,20,21,22,23,24], the current hapten design has better performance in cross-reaction rate and specificity with other triazines. Although the hapten designed and synthesized in this study has only one connecting arm (-CH_2_), it can still stimulate host animals to produce monoclonal antibodies against prometryn with high specificity. This study and the previous hapten design and synthesis of atrazine [18] have confirmed that the isopropyl group of triazine pesticides is also a critical antigenic determinant. Even when there is only one connecting arm (-CH_2_), the active site can still be fully exposed to the immune system. The proposition is that isopropyl is also an antigenic determinant that has crucial guiding significance for the hapten design of triazine pesticides.

## Figures and Tables

**Figure 1 biosensors-13-00022-f001:**
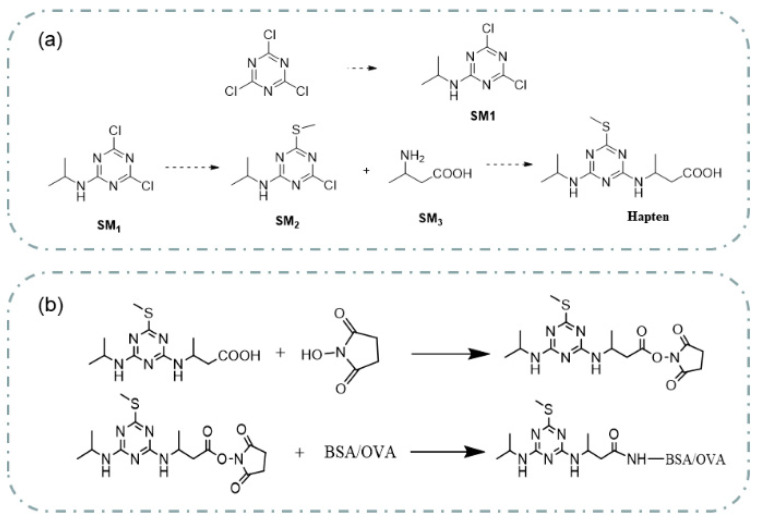
The synthesis route of prometryn hapten (**a**), immunogen and coating antigen (**b**).

**Figure 2 biosensors-13-00022-f002:**
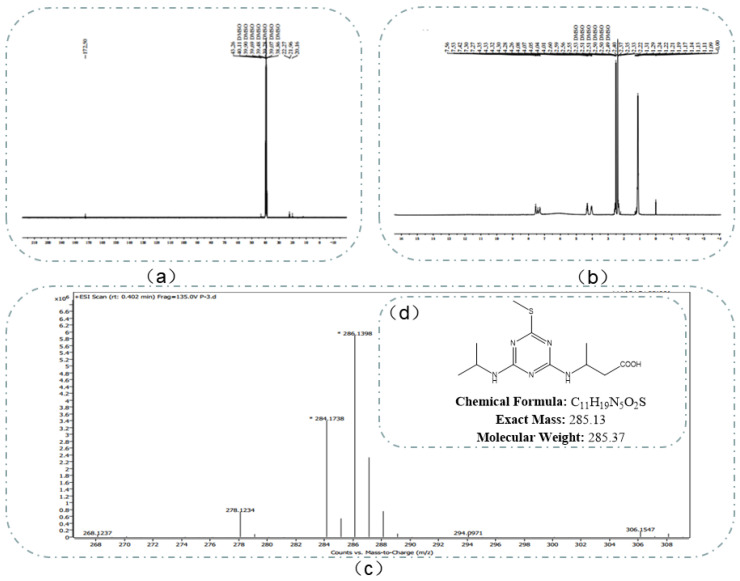
Identification of hapten using ^13^C NMR (**a**), ^1^H NMR (**b**), HRMS (**c**) analyses, and chemical structural formula (**d**).

**Figure 3 biosensors-13-00022-f003:**
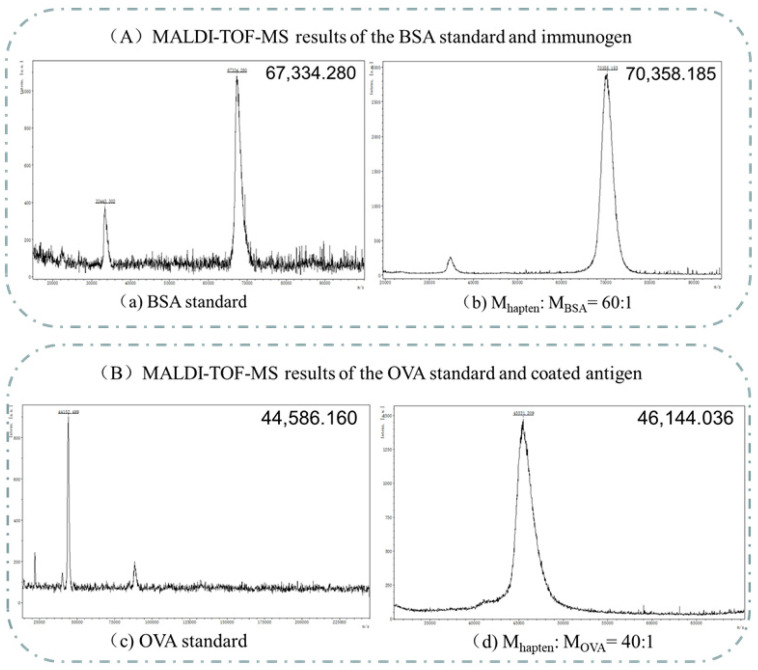
MALDI-TOF-MS results of the immunogen and coating.

**Figure 4 biosensors-13-00022-f004:**
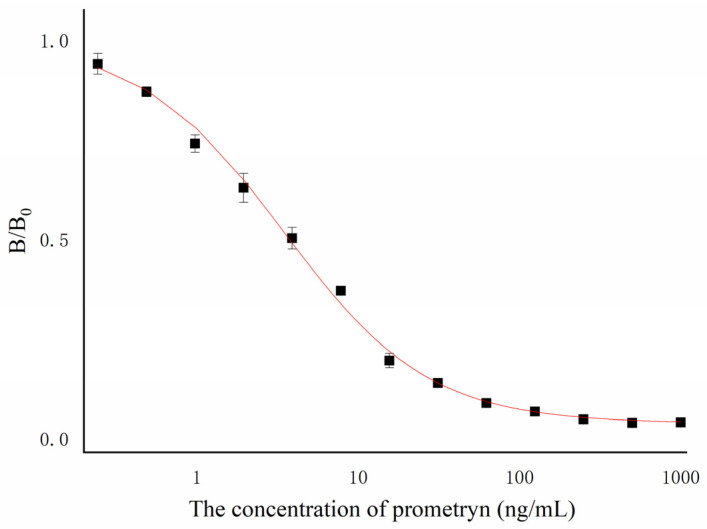
Ic-ELISA standard curve of the 7D4 mAb. (B_0_: OD_450nm_ value of the control hole without inhibitor added; B: OD_450nm_ value of the hole with inhibitor added; B/B0 is called binding rate.)

**Table 1 biosensors-13-00022-t001:** The results of the cross-reactivity of the 7D4 mAb with other triazines.

Triazine Compounds	Chemical Structure	IC_50_ (ng/mL)	Cross-Reactivity
Prometryn	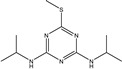	3.901	100%
Ametryn	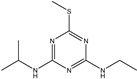	11.221	34.77%
Desmetryn	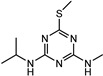	21.566	18.09%
Terbumeton	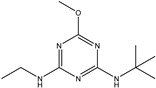	51.091	7.64%
Propazine	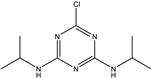	398.011	0.98%
Terbuthylazine	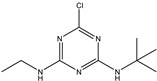	>1000	<0.39%
Simazine	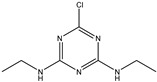	>1000	<0.39%
Simetryn	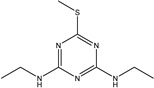	>1000	<0.39%
Atrazine	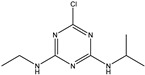	>1000	<0.39%
Prometon	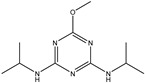	>1000	<0.39%
Terbutryn	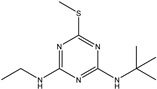	>1000	<0.39%

Note: The concentrations of analyte pesticides were 1000, 500, 250, 125, 62.5, 31.25, 15.625, and 0 ng/mL.

**Table 2 biosensors-13-00022-t002:** The hapten structure of triazine pesticides.

Hapten Name	Hapten Structure	Reference
H1: 4-((4-(isopropylamino)-6-(methylthio)-1,3,5-triazin-2-yl)amino)butanoic acid H2: 3-((4-amino-6-(cyclopropylamino)-1,3,5-triazin-2-yl)thio)propanoic acid	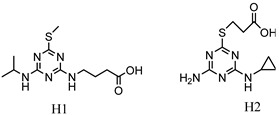	[19]
4-((4-(isopropylamino)-6-(methylthio)-1,3,5-triazin-2-yl)amino)butanoic acid	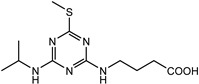	[20,21]
4-chloro-6- (isopropyl amino)-l,3,5-triazine-2-(6-ami- nohexanecarboxylic acid)	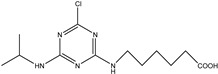	[22,23]
2-mercaptopropionic acid-4-ethylamino-6-isopropylamino-1,3,5-triazine	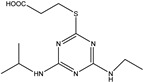	[24]
2-chloro-4-ethylamino-6-isopropylamino-1,3,5-triazine	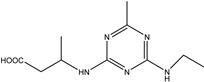	[18]
3-((4-(isopropylamino)-6-(methylthio)-1,3,5-triazin-2-yl)amino)butanoic acid	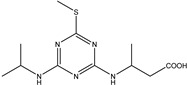	This work

## Data Availability

Not applicable.

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
