# Peer review of "Obtaining a Monoclonal Antibody against a Novel Prometryn-Like Hapten and Characterization of Its Selectivity for Triazine Herbicides"

_biosensors, 2022, doi:10.3390/bios13010022_

Round 1
Reviewer 1 Report
The manuscript entitled "Design and characterization of a novel hapten and preparation of a monoclonal antibody for prometryn detection" is well organized. It is easy to read and understand with a clear description of the experimental design and the results. However, there are some questions that the authors should clarify to improve the quality of this work.
Major:
1. The first part of introduction devotes a lot of space to introduce the background of the study, resulting in the research innovations of this paper not being prominent. It is recommended to compress the content of the introduction section.
2. The author should introduce the development of immunoassays based on antigen antibody specific binding reaction for prometryn detection in the introduction section.
3. Prometryn detection in actual samples should be studied to better illustrate the practicability of this method.
4. Line 319, please rewrite the conclusion part as current conclusion is only experiment result, not a conclusion.
Minor:
1. This manuscript has many problems in English expression and many minor mistakes can be found in this article, which need further consideration and modification.
2. Line 219, 50 L of diluted antiserum? Whether the unit is correct, please check the whole paper.
3. Line 245, change “(NH4)2SO4” into “(NH4)2SO4”, please check the whole paper.
4. The references cited are quite old. Update the references.
5. Your ref citation format is wrong. The page numbers of some refs have not completed. Check all manually.
6. Journal name of the references should be full name or abbreviation? Please check according to the requirement of this journal.
Author Response
Response to Reviewer #1 comments:
The manuscript entitled "Design and characterization of a novel hapten and preparation of a monoclonal antibody for prometryn detection" is well organized. It is easy to read and understand with a clear description of the experimental design and the results. However, there are some questions that the authors should clarify to improve the quality of this work.
Major:
Q1. The first part of introduction devotes a lot of space to introduce the background of the study, resulting in the research innovations of this paper not being prominent. It is recommended to compress the content of the introduction section.
Response: Thank you for the diligent comment. This has been amended as suggested.
Q2. The author should introduce the development of immunoassays based on antigen antibody specific binding reaction for prometryn detection in the introduction section.
Response: Thank you. This has been accomplished as requested.
Q3. Prometryn detection in actual samples should be studied to better illustrate the practicability of this method.
Response: Thank you. Although we agreed with the comment from the diligent reviewer, we cannot afford any further experimental work because of the current situation of COVID, and most of the Universities/Labs are locked.
Q4. Line 319, please rewrite the conclusion part as current conclusion is only experiment result, not a conclusion.
Response: Thank you. This has been done as suggested. Please see the conclusion.
Minor:
Q5. This manuscript has many problems in English expression and many minor mistakes can be found in this article, which need further consideration and modification.
Response: Thank you. The MS has been checked in detail before submission. We confirmed that the MS had been edited by colleagues (among the authorship) who are fluent in English.
Q6. Line 219, 50 L of diluted antiserum? Whether the unit is correct, please check the whole paper.
Response: Thank you. The symbol "μ" has been inserted in the right place throughout the text. Please see Lines 222, 226, 228, 231, and 232.
Q7. Line 245, change “(NH4)2SO4” into “(NH4)2SO4”, please check the whole paper.
Response: Thank you. This has been checked and amended throughout the text. Please see Line 255.
Q8. The references cited are quite old. Update the references.
Response: Thank you. This has been revised according to the useful suggestion. Please see the “References”.
Q9. Your ref citation format is wrong. The page numbers of some refs have not been completed. Check all manually.
Response: Thank you for your kind reminder. This has been revised according to the useful suggestion. Please see the “References”.
Q10. Journal name of the references should be full name or abbreviation? Please check according to the requirement of this journal.
Response: Thank you. This has been revised according to the useful suggestion. Please see the “References”.

Reviewer 2 Report
In this manuscript, the authors have synthesized a new atrazine hapten and used it for the production and isolation of a new monoclonal antibody for detecting atrazine. Then, the isolated antibody was used for the construction of an indirect competitive ELISA (ic-ELISA) method for the determination of atrazine. The novel antibody is with high affinity, specificity, and sensitivity. The IC50 value is 3.901 ng/mL, and low cross-reactivity rates are all characteristics of the produced mAb. It is quite interesting and meaningful to the field of immunoassay. However, some revisions still need to be addressed.
1. Line 54: Please revise “by ingestion of” to “by the ingestion of”, and kindly revise the following text.
2. Line 99: The “cross reaction” should be “cross-reaction”.
3. There should be a blank character between letters and numbers. Please check them all.
4. Line 126: Please kindly revise “Isopropylamine (2.36 g 40 mmoL)” to “Isopropylamine (2.36 g 40 mmol)”.
5. Lines 135 and 134: “with 94.8% yield” and “yellow oil” should be revised to “with a 94.8% yield.” and “a yellow oil”.
6. Line 144: Please kindly correct “a silica gel column (PE:EtOAc 25:1)” to “a silica gel column (PE : EtOAc = 25:1)”.
7. Line 146: Please kindly revise “liquid chromatography‒mass spectrometry (LC‒MS)” to “liquid chromatography-mass spectrometry (LC-MS)”. Dash should be "-".
8. Line 170 and171: Please kindly correct “carrier protein” to “a carrier protein”.
9. Line 184: Please kindly revise the sentence. “The dialysate was changed every 4 h 6 times.” The meaning of this sentence should be “The dialysate shall be replaced every 4 hours for a total of 6 times.”
10. Line 219: Please kindly revise “one hundred microliters” to “100 μL”.
11. Line 239: “Ascites was collected…”, please kindly revise “was” to “were”.
12. Please use the correct significant digits, it’s not necessary to keep such as 0.93746-18.10356 ng/mL for the data. (Lin 327).
13. Some recent references regarding the importance of hapten design for the quality of antibodies against pesticides can be cited. E.g. Huang et al. Science of The Total Environment, 2022, 830, 154690; Chen et al. Journal of Hazardous Materials, 2021, 412, 125241; Chen et al. Ecotoxicology and Environmental Safety, 2020, 196, 110533.
Author Response
Response to reviewer #2’s comments
Manuscript number: biosensors-2016910
Manuscript title: Design and characterization of a novel hapten and preparation of a monoclonal antibody for prometryn detection   Submitted to: biosensor.
Corresponding Authors: Prof. Dr. Maojun Jin, Xiuyuan Zhang.
Dear diligent reviewer,
We would like to thank you and the diligent reviewers for their constructive comments on our manuscript (MS No.: biosensors-2016910). We have considered all the reviewers’ comments, and all amendments are shown in the MS. Without a doubt, the comments significantly improved the quality of MS to meet the high standard of the biosensor.
Herein, the itemized responses to reviewer #2’s comments are as follows:
Please note that all pages and line numbers refer to those in the resubmitted manuscript.
Response to Reviewer #2 comments:
In this manuscript, the authors have synthesized a new atrazine hapten and used it for the production and isolation of a new monoclonal antibody for detecting atrazine. Then, the isolated antibody was used for the construction of an indirect competitive ELISA (ic-ELISA) method for the determination of atrazine. The novel antibody has high affinity, specificity, and sensitivity. The IC50 value is 3.901 ng/mL, and low cross-reactivity rates are all characteristics of the produced mAb. It is quite interesting and meaningful to the field of immunoassay. However, some revisions still need to be addressed.
Q1. Line 54: Please revise “by ingestion of” to “by the ingestion of”, and kindly revise the following text.
Response: Thanks so much for your kind suggestion. This has been done. Please see Line 51.
Q2. Line 99: The “cross reaction” should be “cross-reaction”.
Response: Thank you. This has been done. Please see Line 98.
Q3. There should be a blank character between letters and numbers. Please check them all.
Response: Thank you. This has been done. Please see Lines 128-130.
Q4. Line 126: Please kindly revise “Isopropylamine (2.36 g 40 mmoL)” to “Isopropylamine (2.36 g, 40 mmol)”.
Response: Thank you. This has been done. Please see Lines 128-130.
Q5. Lines 135 and 134: “with 94.8% yield” and “yellow oil” should be revised to “with a 94.8% yield.” and “a yellow oil”.
Response: Thank you. This has been done. Please see Lines 137-138.
Q6. Line 144: Please kindly correct “a silica gel column (PE:EtOAc 25:1)” to “a silica gel column (PE : EtOAc = 25:1)”.
Response: Thank you. This has been done. Please see Line 148.
Q7. Line 146: Please kindly revise “liquid chromatography‒mass spectrometry (LC‒MS)” to “liquid chromatography‒mass spectrometry (LC‒MS)”. Dash should be "-".
Response: Thank you. This has been done.
Q8. Line 170 and171: Please kindly correct “carrier protein” to “a carrier protein”.
Response: Thank you. This has been done. Please see Lines 173 and 174.
Q9. Line 184: Please kindly revise the sentence. “The dialysate was changed every 4 h 6 times.” The meaning of this sentence should be “The dialysate shall be replaced every 4 hours for a total of 6 times.”
Response: Thank you. This has been revised as suggested. Please see Lines 187-188.
Q10. Line 219: Please kindly revise “one hundred microliters” to “100 μL”.
Response: Thank you. This has been revised. Please see Line 222, 228 and 232.
Q11. Line 239: “Ascites was collected…”, please kindly revise “was” to “were”.
Response: Thank you. This has been done. Please see Line 248-249.
Q12. Please use the correct significant digits, it’s not necessary to keep such as 0.93746-18.10356 ng/mL for the data. (Lin 327).
Response: Thank you. This has been provided as suggested. Please see Line 313.
Q13. Some recent references regarding the importance of hapten design for the quality of antibodies against pesticides can be cited. For example, Huang et al. Science of The Total Environment, 2022, 830, 154690; Chen et al. Journal of Hazardous Materials, 2021, 412, 125241; Chen et al. Ecotoxicology and Environmental Safety, 2020, 196, 110533.
Response: Thank you. This has been revised according to the useful suggestion.
Reviewer 3 Report
The submitted article focuses on developing monoclonal antibody to triazine herbicide, prometryn based on novel hapten. The feature of the hapten in comparison with similar one from [19] was to present two symmetrical isopropyl moieties hoping to generate prometryn-selective response. However, the desired selectivity of mAb was not confirmed. The Benefits over previous prometrin immunoassay not clear.
Besides, the work is written carelessly, as evidenced by numerous comments. It had to be carefully checked before submission.
Research doesn't look complete. The paper lacks an application of the developed mAb for determination prometryn in different matrices, recovery experiments, accuracy and precision.
Line 26-28. Since IC20 was 0.9 and IC80 was 18.1. So, change them in the text.
Line 29. High specificity and low cross-reactivity are synonymous. However, mAb demonstrated high cross-reactivity to ametryn and desmetryn 35 and 18%. Therefore, high specificity should be removed.
Line 37. explain where ergometrine is here.
Line 46. Explain to the readers in what units (4.6×106) the carcinogenic risk is measured.
Line 85,86, 98. Prometryn or prometryne. Choose one
Line 94-95 Cross-reactions of 35% and 18% are not low. This level is considered high.
Table 1. Rotate the desmetrin formula clockwise to place the S-CH3 up, as with the rest of the analogs.
Line 101. Trizine?
Line 102 Decipher the abbreviation on first use
Line 124 Clarify the reaction was completed in 15 minutes or 3 h later.
Line 131, 143. Specify SM1, SM2 and SM3.
Line 164-167. This text fragment should be moved to Results section
Line 178. a 4-degree refrigerator change to at 4o C
Line 184. It would be clearer to designate M(conjugate) and M(carrier).
Line 187 Production and characteristics of prometryn mAb should be 3.3 section
Line 195 Add mice to Six 7-week-old female Balb/c …
Line 201. What is fluoride adjuvant?
Line 205-207 Tables are not included in the paper. The results are not available and not discussed.
Line 209. Rabbits were not mentioned in immunization. Remove fusion
Line 221. Specify IgG HRP
Line 222-223 Check 100 L and 50 mL
Line 240. Which antiserum is meant if the section is about mAb production.
Fig 1b. Check and revise amide bond in conjugate. CO-NH
Line 262 Structural formula of prometryne hapten 262 are NOT shown in Figure 2.
C9H14ClN5O2 does not match the hapten formula. Check carefully and revise. Designate hapten in the figure.
Line 274-276. Molar ratios in conjugates are calculated erroneously.
Fig 3 a and b. Correct the molar ratio of hapten:carrier in the synthesis of 60:1 and 40:1 to a hapten load in conjugate 18:1 and 6:1, respectively.
Line 290. Substitute “had an inhibitory concentration” with “was”
Table S3 is not available.
Figure 4. Concentrations below 1 ng/mL must be shown to show the upper asymptote.
IC20 is also not shown on the plot.
Line 295-296. Revise what is IC50
Line 301. What is the principle of IgG1 identification in SDS-PAGE?
Line 314-315. The statements “has very good specificity” and “has low cross reactivity” are not true.
Line 320 A novel structural design of hapten and the resultant antibody specificity should be discussed in comparison with those in [19]. It is also required to justify the need to identify prometrin among other analogues instead of more efficient group detection of most triazines.
Author Response
Response to reviewer #3’s comments
Manuscript number: biosensors-2016910
Manuscript title: Design and characterization of a novel hapten and preparation of a monoclonal antibody for prometryn detection   Submitted to: biosensor.
Corresponding Authors: Prof. Dr. Maojun Jin, Xiuyuan Zhang.
Dear esteemed reviewer,
We would like to thank you and the diligent reviewers for their constructive comments on our manuscript (MS No.: biosensors-2016910). We have considered all the reviewers’ comments, and all amendments are shown in the MS. Without doubt, the comments significantly improved the quality of MS to meet the high standard of the biosensors.
Herein, the itemized responses to reviewer #3’s comments are as follows:
Please note that all pages and line numbers refer to those in the resubmitted manuscript.
Response to Reviewer #3 comments:
The submitted article focuses on developing a monoclonal antibody to the triazine herbicide prometryn based on a novel hapten. The feature of the hapten in comparison with a similar one from [19] was to present two symmetrical isopropyl moieties hoping to generate a prometryn-selective response. However, the desired selectivity of mAb was not confirmed. The benefits over previous prometrin immunoassays are not clear.
Response: Thank you so much for your kind suggestion. The ic-ELISA immunogen established in Reference 19 is a compound with an H1 structure, and the coating is a compound with an H2 structure. Immunogen (H1 – OVA) and coating antigen (H2 – BSA) are used to determine the antibody performance when producing antibodies. The hapten synthesis path of Reference 19 is shown in the figure. The ic-ELISA method established in this study is a homologous ic-ELISA, that is, the coating antigen is the same as the hapten used by the immunogen. The hapten of prometryn designed and synthesized in this study only has one carbon link arm(-CH2) and active group (-COOH). This study is not intended to be compared with reference 19 of the previous study. The main purpose of this study is to fully prove that the isopropyl group of prometryne is an antigen determinant. Xu, the first author, designed and synthesized the hapten of atrazine and prepared the monoclonal antibody against atrazine. The hapten determinant of triazine pesticides is isopropyl. According to the hapten design principle, even if the connecting arm is one and the active group, it is still used to prepare highly sensitive and specific triazine monoclonal antibodies.
In addition, the work is written carelessly, as evidenced by numerous comments. It had to be carefully checked before submission.
Response: Thank you for your kind reminder. Our apologies for the carelessness have been revised. MS has been checked in detail before submission. We confirmed that the MS had been edited by colleagues (among the authorship) who are fluent in English.
Research does not look complete. The paper lacks an application of the developed mAb for determination prometryn in different matrices, recovery experiments, accuracy and precision.
Response: Thank you for the instructive comment. We agree with the diligent reviewer; however, we cannot afford any further experimental work because of the current situation of COVID, and most of the Universities/Labs are locked.
Q1. Line 26-28. The IC20 was 0.9, and the IC80 was 18.1. Therefore, change them in the text.
Response: Thank you for the kind advice. The revised text is as follows.
“The IC20 (concentration of analyte causing 80% inhibition of prometryn)-IC80 (concentration of analyte producing 20% inhibition of prometryn) range of the anti-prometryn monoclonal antibody's standard curve's working range was 0.9-18.1 ng/mL.”
Q2. Line 29. High specificity and low cross-reactivity are synonymous. However, mAb demonstrated high cross-reactivity to ametryn and desmetryn 35 and 18%, respectively. Therefore, high specificity should be removed.
Response: Thank you so much for your kind suggestion. This has been done. Please see Line 29.
Q3. Line 37. explain where ergometrine is here.
Response: Thank you for the diligent comment. An abstract of Reference 2 is presented below.
Uniformly 14C-ring-labeled prometryn [2-(methylthio)-4,6-bis(isopropylamino)-s-triazine] was incubated with organic soil under laboratory conditions for 1 year. After exhaustive solvent extraction, the soil containing bound (nonextractable) 14C-labeled residues was fractionated into humic substances by alkali extraction followed by acid precipitation. A considerable proportion of 14C-labeled residues in humin and humic acid comprised the parent herbicide and its mono-N-dealkylated metabolite. However, the soluble fulvic acid fraction contained an appreciable amount of the 2-hydroxy analog of prometryn. Thermoanalytical methods were used to obtain information on the nature and location of 14C-labeled bound residues in soil and humic materials. Exhaustive methylation of humin and humic acid released some of the bound "C-labeled residues from these fractions. Bound 1"C-labeled residues in aqueous suspensions of human or humic acid were stable to UV light. However, UV irradiation of the soi suspension resulted in the release of some of the bound 14C-labeled residues, which were subsequently decomposed to hydroxy analogs.
Q4. Line 46. Explain to the readers in what units (4.6×106) the carcinogenic risk is measured.
Response: Thank you for the kind advice. The original expression of Reference 5 is as follows.
“Alachlor, atrazine, and a-HCH showed unacceptable carcinogenic risk estimates (4.5E-06,4.6E-06 and 1.3E-04, respectively).” No units are listed in reference 5.
Q5. Line 85,86, 98. Prometryn or prometryne. Choose one
Response: Thank you for your kind comment. “Prometryn” was selected.
Q6. Line 94-95 Cross-reactions of 35% and 18% are not low. This level is considered high.
Response: Thank you for the kind suggestion. The revised text according to the suggestions is as follows.
“The cross-reaction rate of the prepared prometryn mAb with other triazines is low, except ametryn and desmetryn.”
Q7. Table 1. Rotate the desmetrin formula clockwise to place the S-CH3 up, as with the rest of the analogs.
Response: Thank you so much for your kind suggestion. It has been revised according to the suggestions, as shown in the figure.
Q8. Line 101. Trizine? 三嗪?
Response: Thank you for the kind advice. To make the reader clearer, it has been revised as follows.
“The trizine standard (prometryn, ametryn, desmetryn, terbumeton, propazine, terbuthylazine, simazine, simetryn, atrazine, prometon, terbutryn) used by the Institute was secured from the first standard (Tianjin, China).”
Q9. Line 102 Decipher the abbreviation on first use.
Response: Thank you for the kind suggestion. The full name of DIEPA is N,N-diisopropylethylamine. We have confirmed the abbreviations and revised them as follows in the submitted version.
Q10. Line 124 Clarify the reaction was completed in 15 minutes or 3 h later.
Response: Thank you for the diligent comment.
“The process of dropping solution 2 into solution 1 takes approximately 15 minutes to complete. Then, the mixture was reacted at 0 ℃ for 3 h, and thin layer chromatography (TLC) was used to check whether the reaction was completed.”
Q11. Line 131, 143. Specify SM1, SM2 and SM3.
Response: Thank you for the diligent comment. SM1: 4,6-dichloro-N-isopropyl-1,3,5-triazin-2-amine; SM2: 4-chloro-N-isopropyl-6-(methylthio)-1,3,5-triazin-2-amine; SM3: 3-aminobutanoic acid.
Q12. Line 164-167. This text fragment should be moved to Results section
Response: Thank you for the kind advice. This has been revised according to the useful suggestion. Please see section 5.1.1.
Q13. Line 178. a 4-degree refrigerator change to at 4℃.
Response: Thank you for your kind reminder. This has been revised according to the useful suggestion.
Q14. Line 184. It would be clearer to designate M(conjugate) and M(carrier).
Response: Thank you for the instructive comment. This has been revised according to the useful suggestion.
Q15. Line 187 Production and characteristics of prometryn mAb should be 3.3 section
Response: Thank you for the instructive comment. This has been revised according to the useful suggestion.
Q16. Line 195 Add mice to Six 7-week-old female Balb/c …
Response: Thank you for the instructive comment. This has been revised according to the useful suggestion.
Q17. Line 201. What is fluoride adjuvant?
Response: Thank you for your kind comment. Our apologies for the carelessness; this has been revised. It has been revised to “Freund's”.
Q18. Line 205-207 Tables are not included in the paper. The results are not available and not discussed.
Response: Thank you for the kind advice. Line 205-207 Tables are in the supplementary materials. Please see the supplementary materials. Table data show the serum titer and inhibition rate of prometryne fusion mice.
Q19. Line 209. Rabbits were not mentioned in immunization. Remove fusion.
Response: Thank you for the instructive comment. This has been revised according to the useful suggestion.
Q20. Line 221. Specify IgG HRP
Response: Thank you so much for your kind suggestion. IgG HRP was purchased from Jackson Immuno. Research Laboratories Inc., Pennsylvania, USA.
Q21. Line 222-223 Check 100 L and 50 mL
Response: Thank you for your kind reminder. The symbol "μ" is not displayed after the uploaded Word version is converted into the template word of the journal.
Q22. Line 240. Which antiserum is meant if the section is about mAb production.
Response: Thank you so much for your kind suggestion. This part has been revised in the MS. During the preparation of the monoclonal antibody, this part should be ascites produced by the hybridoma cell line in the abdominal cavity of mice. Antiserum generally refers to mouse polyclonal antiserum or rabbit polyclonal antiserum.
Q23. Fig 1b. Check and revise amide bond in conjugate. CO-NH
Response: Thank you for the instructive comment. This has been revised according to the useful suggestion. Please see Fig 1b.
Q24. Line 262 Structural formula of prometryne hapten 262 are NOT shown in Figure 2.
Response: Thank you for your kind comment. The determination results of the chemical structural formula of the prometryne hapten are shown in Figure 2(d).
Q25. C9H14ClN5O2 does not match the hapten formula. Check carefully and revise. Designate hapten in the figure.
Response: Thank you. Our apologies for the carelessness; this has been revised. The hapten formula is C11H19N5O2S. The determination results of the chemical structural formula of the prometryne hapten are shown in Figure 2(d).
Q26. Line 274-276. Molar ratios in conjugates are calculated erroneously.
Response: Thank you for the diligent comment. Our apologies for the carelessness; this has been revised.
Conjugation ratio = (Mconjugate − Mcarrier)/Mhapten.
70358.185-67334.280/285.37=10(keep integer)
46144.036-44586.160/285.37=5(keep integer)
Q27. Fig 3 a and b. Correcting the molar ratio of hapten:carrier in the synthesis of 60:1 and 40:1 to hapten loads in conjugates 18:1 and 6:1, respectively.
Response: Thank you for your kind comment. Our apologies for the carelessness; this has been revised. Correct the molar ratio of hapten:carrier in the synthesis of 60:1 and 40:1 to hapten loads in conjugates of 10:1 and 5:1, respectively.
Q28. Line 290. Substitute “had an inhibitory concentration” with “was”.
Response: Thank you so much for your kind suggestion. This has been done.
Q29. Table S3 is not available.
Response: Thank you for your kind comment. Please see the supplementary materials. Table S3 shows the optimal combination of antigen and antibody. The optimal combination of antigen and antibody with an OD450 nm value of approximately 1.0 was selected to ensure the accurate determination of the OD450 nm value. Finding the optimal combination of antigen and antibody has a great impact on the establishment of the antibody standard curve, so Table S3 is necessary.
Q30. Figure 4. Concentrations below 1 ng/mL must be shown to show the upper asymptote.
Response: Thank you for the instructive comment. The abscissa axis of the standard curve was adjusted to less than 1 ng/mL.
Q31. IC20 is also not shown on the plot.
Response: Thank you for the instructive comment. The result value of the fitted curve is the value that can display IC20.
Q32. Line 295-296. Revise what is IC50?
Response: Thank you for the kind advice. This has been revised according to the useful suggestion. The IC50 was the quantity of analyte that generated 50% prometryn inhibition.
Q33. Line 301. What is the principle of IgG1 identification in SDS‒PAGE?
Response: Thank you. Our apologies for the carelessness; this has been revised. The 7D4 mAb type was identified as IgG1 using a mouse antibody isotyping kit from Sigma®, St. Louis, MO, USA.
Q34. Line 314-315. The statements “has very good specificity” and “has low cross reactivity” are not true.
Response: Thank you for the instructive comment. This has been revised according to the useful suggestion. Relevant expressions have been deleted.
Q35. Line 320 A novel structural design of hapten and the resultant antibody specificity should be discussed in comparison with those in [19]. It is also required to justify the need to identify prometrin among other analogs instead of more efficient group detection of most triazines.
Response: Thank you for the instructive comment. This has been revised according to the useful suggestion. Relevant expressions have been deleted. The conclusion has been revised. Please see section 5.

Round 2
Reviewer 1 Report
The authors have made proper revisions.
Author Response
Thank you. We tried our best to fulfil all queries raised by the diligent reviewer.
Reviewer 3 Report
The revision of the manuscript addressed only some errors and typos, leaving the rest unchanged.
The authors still did not indicate what is the shortcoming of the described triazine immunoassays and what is the need to improve them. What problem did the present study solve?
The authors have added an important table 2 including a variety of triazine haptens for producing anti-promethrin antibodies. However, the characteristics of these reported assays (sensitivity and cross-reactivity) are not presented, which does not allow evaluating and proving the benefits of the new hapten.
The work still remained without confirmation of the practical use of the developed mAb-based test. This does not allow us to consider the study completed.
Comments that have not changed are repeated below.
L29. The authors confuse the generally accepted concepts of binding level and inhibition rate. IC is inhibition concentration, IC20 is analyte concentration resulting 20% binding inhibition or 80% binding level (B/Bo). And not vice versa.
L31 “High affinity, sensitivity, and low cross-reactivity rates are all characteristics of the produced mAb”. These characteristics are not supported by any data.
L40 Ergometrine is a medicine used in obstetrics. It has nothing to do with triazines and, it is not mentioned in [2], as the authors are sure,
L 217 What is “fusion mouse sera”? Is it mice taken for spleen cell fusion OR pooled sera
L228 What is IgG HRP?
Figure 1b. The structures of intermediates and conjugates remain incorrect. Compare with your previous paper [18].
Line 225. Rabbits were not mentioned in immunization. Remove it or provide immunization of rabbits
Fig 3 a and b. MALDI spectra confirmed hapten load. So revise molar ratio to 10:1 and 5:1, respectively.
Figure 4. Concentrations of analyte should be taken <1 ng/mL to show the whole range of calibration curve including the upper asymptote.
L309 Revise. IC50 is concentration not quantity.
L 342 “There is no doubt that the hapten designed and synthesized in this study has good specificity” is nonsensical phrase.
Author Response
Response to reviewer #3’s comments
Manuscript number: biosensors-2016910
Manuscript title: Design and characterization of a novel hapten and preparation of a monoclonal antibody for prometryn detection   Submitted to biosensor.
Corresponding Authors: Prof. Dr Maojun Jin, Xiuyuan Zhang.
Dear esteemed reviewer,
We want to thank you for your constructive comments on our manuscript (MS No.: biosensors-2016910). We have considered all the comments, and all amendments are shown in red highlights. Without a doubt, the comments substantially improved the quality of MS to meet the high standard of biosensors.
Herein, the itemized responses to reviewer #3’s comments are as follows:
Please note that all pages and line numbers refer to those in the resubmitted manuscript.
Response to Reviewer #3 comments:
Q1. The revision of the manuscript addressed only some errors and typos, leaving the rest unchanged.
Response: Thank you. We tried our best to fulfil all queries raised by the diligent reviewer.
Q2. The authors still did not indicate the shortcoming of the described triazine immunoassays and the need to improve them. What problem did the present study solve?
Response: Thank you for the instructive comment.
- There are two problems with the previous triazine immunoassays. First, there are few reports on haptens specially designed for prometryn. Second, the cross-reaction rate of antibodies prepared by the previously reported hapten was high.
- The necessity of this study has two points. First, there are a few controversial reports on the design of haptens, specifically recognizing prometryn and the preparation of monoclonal antibodies. Second, the hapten of prometryn designed and synthesized in this study ultimately competes with the design of prometryn, and this study proposed the isopropyl group, one of the most critical antigenic determinants of triazine pesticide hapten design. Antibiotics are undoubtedly the core element for establishing immunoassay methods, so this study proposes a novel design and synthetic path for prometryn haptens.
- There are two problems to be solved in this study. On the one hand, the previously reported triazine hapten competes for the hapten design of prometryn. On the other hand, this study is the first report on the specific preparation of monoclonal antibodies against prometryn, which solves the design and synthesis of prometryn hapten.
The problem of cross-rate reaction between triazine pesticides previously reported was solved.
Response: Thank you for the diligent comment.
Q3. The authors have added an important table 2, including a variety of triazine haptens for producing anti-promethrin antibodies. However, the characteristics of these reported assays (sensitivity and cross-reactivity) are not presented, which does not allow evaluating and proving the benefits of the new hapten.
Response: Thank you. This has been clearly stated in Reference 18.
Q4. The work still remained without confirmation of the practical use of the developed mAb-based test. This does not allow us to consider the study completed. Comments that have not changed are repeated below.
Response: Thank you. That is because of POST-COVID shortcomings.
Q5. L29. The authors confuse the generally accepted concepts of binding level and inhibition rate. IC is the inhibition concentration, and IC20 is the analyte concentration resulting in 20% binding inhibition or 80% binding level (B/Bo). In addition, not vice versa.
Response: Thank you. This has been amended as raised by the esteemed reviewer Line 29.
Q6. L31 “High affinity, sensitivity, and low cross-reactivity rates are all characteristics of the produced mAb”. These characteristics are not supported by any data.
Response: Thank you. This has been revised. Please see Line 29.
Q7. L40 Ergometrine is a medicine used in obstetrics. It has nothing to do with triazines and, it is not mentioned in [2], as the authors are sure,
Response: Thank you for the kind advice.
Q8. L 217 What is “fusion mouse sera”? Is it mice taken for spleen cell fusion OR pooled sera.
Response: Thank you. Only one of the six immunized mice was selected for the fusion test after the serum titer was determined. The selected mice are called fusion mice. The serum of selected mice collected is called the serum of fusion mice.
Q9. L228 What is IgG HRP?
Response: Thank you. IgG-HRP: label goat anti-mouse IgG with HRP. The full name of IgG HRP is goat anti-mouse IgG Fc (HRP), Conjugation: HRP, Host species: Goat, Isotype: IgG.
Q10. Figure 1b. The structures of intermediates and conjugates remain incorrect. Compare with your previous paper [18].
Response: Thank you. It is genuinely different. Because Reference 18 was designed for atrazine and this study was designed for prometryn, the intermediates of these two studies are different.
Q11. Line 225. Rabbits were not mentioned in immunization. Remove it or provide immunization of rabbits
Response: Thank you. This has been revised as suggested. Please see Line 227.
Q12. Fig 3 a and b. MALDI spectra confirmed hapten load. So revise molar ratio to 10:1 and 5:1, respectively.
Response: Thank you. This has been revised according to the valuable suggestion. The molar ratio is shown in Fig. 3a, and b show hapten and carrier protein input molar ratio during complete antigen synthesis. Therefore, the molar ratio in Fig. 3a and b does not need to be revised. The coupling ratio has been corrected in the text in the last revision. Please see Line 295.
Q13. Figure 4. Concentrations of analyte should be taken <1 ng/mL to show the whole range of the calibration curve, including the upper asymptote.
Response: Thank you. This has been revised as suggested. Please see Figure 4.
Q14. L309 Revise. IC50 is concentration not quantity.
Response: Thank you. This has been amended as suggested. Please see Line 310.
Q14. L 342“There is no doubt that the hapten designed and synthesized in this study has good specificity” is a nonsensical phrase.
Response: Thank you. This sentence has been deleted.

Round 3
Reviewer 3 Report
Most remarks remain the same
Author Response
Thank you for the diligent comment.